# Cell-Free DNA Fragmentomics: The Novel Promising Biomarker

**DOI:** 10.3390/ijms24021503

**Published:** 2023-01-12

**Authors:** Ting Qi, Min Pan, Huajuan Shi, Liangying Wang, Yunfei Bai, Qinyu Ge

**Affiliations:** 1State Key Laboratory of Bioelectronics, School of Biological Science and Medical Engineering, Southeast University, Nanjing 210096, China; 230228564@seu.edu.cn (T.Q.); 18705168526@163.com (H.S.); 220214918@seu.edu.cn (L.W.); whitecf@seu.edu.cn (Y.B.); 2School of Medicine, Southeast University, Nanjing 210097, China; panmin1120@163.com

**Keywords:** cell-free DNA, fragmentomics, biomarker, applications, tissue-of-origin

## Abstract

Cell-free DNA molecules are released into the plasma via apoptotic or necrotic events and active release mechanisms, which carry the genetic and epigenetic information of its origin tissues. However, cfDNA is the mixture of various cell fragments, and the efficient enrichment of cfDNA fragments with diagnostic value remains a great challenge for application in the clinical setting. Evidence from recent years shows that cfDNA fragmentomics’ characteristics differ in normal and diseased individuals without the need to distinguish the source of the cfDNA fragments, which makes it a promising novel biomarker. Moreover, cfDNA fragmentomics can identify tissue origins by inferring epigenetic information. Thus, further insights into the fragmentomics of plasma cfDNA shed light on the origin and fragmentation mechanisms of cfDNA during physiological and pathological processes in diseases and enhance our ability to take the advantage of plasma cfDNA as a molecular diagnostic tool. In this review, we focus on the cfDNA fragment characteristics and its potential application, such as fragment length, end motifs, jagged ends, preferred end coordinates, as well as nucleosome footprints, open chromatin region, and gene expression inferred by the cfDNA fragmentation pattern across the genome. Furthermore, we summarize the methods for deducing the tissue of origin by cfDNA fragmentomics.

## 1. Introduction

In 1948, circulating free nucleic acids (cell-free DNA) were first identified in normal individuals and diseased populations [1]. Later, the presence of circulating free DNA was found successively in cancer patients [2], SLE patients [3], and organ transplant recipients [4]. In 1997, Dennis Lo demonstrated the presence of fetal DNA in the plasma and serum of pregnant women for the first time [5]. Cell-free DNA (cfDNA) is the free fragment of DNA molecules in plasma derived from apoptotic and necrotic cells with different topologies [6], and circulating tumor DNA (ctDNA) is derived from tumor cells [7]. Further, cfDNA has the potential to be applied in noninvasive prenatal diagnosis [8], monogenic diseases [9], cancer [10], and minimal residual diseases [11]. However, the application of plasma DNA in diagnosing cancer and fetus diseases still faces many difficulties due to the inability to distinguish the normal-derived cfDNA and tumor-derived or fetal-derived cfDNA.

CfDNA is considered as cellular waste, but the information characterized by cfDNA is an important biomarker reflecting the physiological state [12,13]. The characterization of cfDNA fragments was mainly focused on the fragment length distribution [14] and fragment integrity [15]. It was shown that ctDNA shows higher fragmentation [16], thus ctDNA can be identified by enriching short molecules [17]. The identification of shorter ctDNA facilitates the noninvasive detection and monitoring of cancer [18,19,20]. However, there exists a high probability of false positives in defining ctDNA by detecting mutations carried by cfDNA, because the majority of variants carried by plasma DNA are derived from clonal hematopoiesis in leukocytes. Moreover, tumor-derived cfDNA may not carry variants [21]. Therefore, probing the fragment characteristics of cfDNA at the genome-wide level can provide a more effective approach for biomarker mining, as tumor-derived cfDNA harbors different fragment characteristics compared with normal-cell-derived cfDNA [22] (Figure 1B,C).

In 2015, fragmentomics was first introduced by Ivanov et al., whose study showed that the non-random fragmentation pattern of cfDNA reflects epigenetic regulation [23]. Further, it was shown that the fragmentation of cfDNA comprises an in vivo nucleosome footprint, which can inform the tissue of origin [24]. In 2019, Cristiano et al. discovered that the degree of cfDNA fragmentation differs across the genome and developed a method to evaluate the pattern of cfDNA fragmentation across the genome, finding different fragmentation profiles in normal individuals and individuals with cancer [25]. Studies have indicated that the cleavage of cfDNA by nucleases is tendentious [26,27], and characteristics such as the end motifs [28] (Figure 1C) and preferred end coordinates [29] (Figure 2) in cfDNA molecules may support this view. In the last two years, some studies have integrated fragmentomics features, including fragment length, fragment size distribution, end motifs, preferred end coordinates, and breakpoint motifs, to sensitively detect cancers [30,31,32]. However, the tiny amount of cfDNA leads to problems in library construction, and the limited screening accuracy in different states restricts its application in the early diagnosis of tumors and abnormal pregnancies.

This review summarizes the characteristics of cfDNA fragmentomics and its potential applications, such as fragment length, end motifs, jagged ends, breakpoint coordinates, and motifs, as well as nucleosome footprints, open chromatin region, and gene expression characterized by the distribution pattern of cfDNA fragments across the genome. Moreover, we list the methods to analyze and identify the tissue of origin from the cfDNA fragmentomics (Table 1), expecting to provide a theoretical basis for novel biomarker mining from the perspective of cfDNA fragmentomics.

## 2. Fragment Size of cfDNA

### 2.1. Enhanced Disease Detection by Enriching Short cfDNA

The length of cfDNA fragments harbors the potential to reflect the physiological state of an individual. Earlier, differences in cfDNA fragment length were mainly used to isolate circulating tumor DNA [37] and fetal free DNA [38]. Shi et al. suggested that cfDNA fragment size profiles shared common features in different populations and were distributed as 10 bp-step patterns [39]. The length of cfDNA from tumor cells (dominant peak at ~143 bp) is shorter than that from normal cells (dominant peak at ~167 bp) [40] (Figure 1B). Similarly, the length of cfDNA from placental cells (dominant peak at ~146 bp) is shorter than that from maternal cells (dominant peak at ~166 bp) [41]. The underlying mechanism for this phenomenon may be related to the cleavage sites of nucleases. Fetal DNA was frequently cleaved within the nucleosome core, while maternal DNA was mostly cleaved within the linker region [42], which explains the molecular biological mechanism of fragment length differences.

Based on the differences in overall cfDNA fragment size patterns, many related applications focused on the short cfDNA fragments have emerged. In 2014, Yu et al. found that the proportion of short fragments of cfDNA in fetal chromosomal aneuploidy was aberrant, by which they can detect fetal trisomy 21 and trisomy 18 with 100% sensitivity [43]. Moreover, Qiao et al. improved fetal fraction in noninvasive prenatal diagnosis by enriching shorter cfDNA fragments [44]. Similarly, Liu’s team improved the detection of pancreatic cancer by enriching for short mutant cfDNA fragments [17]. Several studies have been conducted to investigate the level of short fragments in cfDNA [45,46]. However, the distribution pattern of short fragments varies with individual physiological status and among sample batches.

Exploring the distribution of fragment lengths of cfDNA is important for mining potential cfDNA biomarkers. Yu et al. showed that fetal DNA fractions can be deduced from the overall size distribution of maternal plasma DNA [43]. Arbabi et al. relied on the fragment size distribution of cfDNA in samples to identify sub-chromosomal copy number variants in the fetal genome [47]. Mouliere et al. first obtained the distribution pattern of cfDNA fragments in tumor patients through the deep sequencing of cfDNA and developed a fragment size selection method by enriching the fragment length by 90–150 bp, which can achieve an enhanced detection of cancer by integrating the fragment length and copy number variants [33]. Moreover, a non-negative matrix factorization (NMF) algorithm has been used to infer the true tumor fragment length distribution correlates with ctDNA level [48]. Nevertheless, the fragment distribution patterns of cfDNA vary among individuals [49,50]; further work is required to quantify the degree of variability this technology could have before it is applied to clinical testing.

### 2.2. The Presence of Ultrashort, Ultralong and Circular cfDNA

Since attention was previously mainly on double-stranded linear plasma DNA, ultralong double-stranded DNA, ultrashort single-stranded cfDNA, and circular cfDNA were ignored. In addition to 100–300 bp of cfDNA, Jahr et al. observed DNA molecules longer than 10,000 bp using polyacrylamide gel electrophoresis [6]. Recently, the team of Dennis Lo revealed a large number of long fragments (fragment length > 500 bp) of cfDNA in maternal plasma using single-molecule sequencing technology [51]. In 2022, their team also examined long cfDNA in cancer patients using SMRT-seq and found the longest tumor cfDNA was 13.6 kb [52]. In addition to the discovery of long cfDNA, Hudecova et al. identified a large number of single-stranded cfDNA of about 50 bp using the combination of high-affinity magnetic bead-based DNA extraction and single-stranded DNA sequencing library preparation (MB-ssDNA). Furthermore, they found the abundance of ultrashort cfDNA in the cancer patients is lower compared to that in the normal population. Moreover, their study implied that cfDNA can characterize the secondary structures (G4 structures) of gene regulatory regions [53]. In addition to linear cfDNA, Sin et al. explored the presence of extrachromosomal circular DNA (eccDNA) in the plasma of pregnant women. Moreover, fetal-derived eccDNAs were shorter than the maternal-derived eccDNAs [54]. The in-depth study of cfDNA with different fragments and different topologies contributes to a more systematic and comprehensive understanding of cfDNA, which provides the basis for further application.

## 3. End Characteristics of cfDNA

### 3.1. The Promising Biomarkers: cfDNA End Characteristics

The ends of cfDNA have regular characteristics, including end motifs (Figure 1A), jagged ends (Figure 1A), and preferred end coordinates (Figure 2). Unlike the fragment length, the sequence characteristics of cfDNA have not received attention until recent years. In 2016, Chan et al. demonstrated that a subset of human genomic locations is preferentially cleaved when plasma DNA molecules are generated, called plasma DNA preferred ends [55] (Figure 2). Furthermore, preferred ends are selective for fetus-derived or maternal-derived cfDNA, which is correlated with the fetal fraction. More recently, Jiang et al. showed that there exists a class of ctDNA signature in the form of preferred end coordinates, and the abundance of preferred ends correlates with the tumor fraction of hepatocellular carcinoma [29]. In addition, long and short fragments of cfDNA were proved to have different preferred ends in maternal peripheral blood. Thus, the featured end of cfDNA in body fluid impels the application potential of cfDNA characteristics in the early diagnosis of cancer and abnormal pregnancies.

Later, Jiang et al. analyzed four bases in 5′ and 3′ of plasma DNA fragments using massive parallel sequencing and found that the most frequently occurring end motif (CCCA) was less abundant in patients with HCC than in healthy subjects (Figure 1C). Moreover, the diversity of cfDNA end motifs significantly increased in cancer patients [28]. Inspired by end motifs, several studies utilized upstream and downstream nucleotides of cfDNA fragment break points (called breakpoint motifs) to predict early lung cancer [56]. Therefore, the profile of end motifs may serve as a class of biomarkers for liquid biopsy. Moreover, plasma DNA originating from different tissues harbor various end motif characteristics, which can reveal the tissue of origin of cfDNA.

In 2020, the group of Dennis Lo examined the jagged ends of cfDNA by identifying methylation signatures [57] and found the proportion of jagged ends was higher in ctDNA and cffDNA (cell-free fetal DNA). However, the ratio of jagged ends in urine cfDNA is distinct to that in plasma cfDNA. It was reported that the jagged end cfDNAs in urine are mainly enriched in the nucleosome linker region, which is more susceptible to degradation owing to the lack of histone protection [58]. This leads to the conclusion that cfDNA is more fragmented in urine [59]. Additionally, the specific length of the jagged ends can be determined based on the sequencing analysis using high-resolution jag-seq. Meanwhile, Xie et al. found that the jagged end length of cfDNA in urine was longer than that in plasma [60].

### 3.2. Effect of Nucleases on End Characteristics

The formation of cfDNA is inextricably linked to the mechanism of nucleases [27]. Serpas et al. found that the deletion of the Dnase1l3 gene causes changes in cfDNA length and end motif frequency [61]. This result suggests that the appearance of cfDNA end motifs may have a preference due to the type of nucleases. Moreover, a previous study has suggested that aberrations in the profile of plasma DNA jagged ends correlates with the type of nuclease that was genetically deleted, depending on nucleosome structures [62]. For example, the deletion of Dnase1l3 led to a significant reduction in jaggedness for those plasma DNA molecules involving more than one nucleosome [62]. Hence, detailed knowledge of the relationship between nucleases and plasma DNA opens up the way for the explanation of cfDNA characteristics.

## 4. Fragmentation of cfDNA

Related studies have shown that the fragmentation pattern of cfDNA is distributed non-randomly across the genome [23]. Cristiano et al. developed an approach (DELFI) to evaluate fragmentation patterns across the genome and found that the fragment profiles of patients with cancer are more varied compared to healthy individuals [25,63]. Based on these findings, epigenetic features, such as nucleosome footprints and gene expression, can be inferred according to the fragmentation pattern of cfDNA [23] (Table 1).

### 4.1. Inference on Nucleosome Footprints

For closed chromatin, DNA is wrapped in histones protected from nuclease degradation. Thus, the cfDNA fragmentation pattern has the potential to reflect nucleosome positioning (Figure 2). Fan et al. found cfDNA fragments enriched in nucleosome sequences while detecting polyploidy using the shotgun method [8]. Nucleosome footprints differ in cells. Girirajan et al. generated maps of genome-wide nucleosome occupancy in vivo and found that short cfDNA fragments harbor the footprints of transcription factors [24] (Figure 2). Furthermore, the cfDNA nucleosome occupancy correlates well with the epigenetic regulation in cells, suggesting that they could inform the cells’ tissue of origin [24]. Moreover, an analysis of urine cfDNA fragmentation patterns also revealed conserved regions in the genome that were repeatedly protected, and the protected regions were partially overlapped with the nucleosome footprints inferred by plasma cfDNA [59]. In brief, nucleosome footprints can be used to inform the cell types contributing to the cfDNA pool in pathological states such as cancer or pregnancy, which makes it possible to identify the tissue origin of cancer cells noninvasively.

So far, several applications of nucleosome occupancy of cfDNA have emerged. In 2016, research on noninvasive prenatal diagnosis demonstrated that fetal fractions could be estimated based on genome-wide nucleosome footprints [64]. Chen et al. took nucleosome occupancy as one of the important features in the logistic regression model for the early detection of hepatocellular carcinoma in patients with cirrhosis [65]. In addition, transcript factor (TF) activity can be evaluated by the nucleosome occupancy of cfDNA [66]. With the increasing amount of sequencing data of cfDNA, bioinformatics researchers collect and organize cfDNA fragment data into relevant databases [67,68], facilitating data access for interested parties. Their work provides an effective resource platform for the development of liquid biopsy-based biomarkers.

### 4.2. Inference on Gene Expression

Gene expression is associated with epigenetic regulation [69]. The actively expressed genes are located in an open chromatin region flanked by well-positioned nucleosomes (Figure 2). In this open chromatin region, approximately 150 bp upstream and 50 bp downstream of the promoter is the nucleosome-depleted region (NDR), where the coverage of cfDNA fragments is lower than the surrounding region [70,71]. Ulz et al. identified two discrete regions in the transcription start site (TSS) using whole-genome sequencing of cfDNA, where nucleosome occupancy resulted in different coverage depths of the expressed and silenced genes [72]. Their study indicates that the expressed genes could be inferred from the cfDNA coverage depth pattern of the TSS. Based on their discovery, Yang’s team applied it to noninvasive prenatal diagnosis for predicting pregnancy complications. They utilized cfDNA to infer expressed genes in pregnant women to distinguish pregnant women with pregnancy complications and healthy women [73,74]. Currently, the application of noninvasive inference of expressed genes by fragment coverage patterns of cfDNA in the TSS is relatively rare, since a good experimental result requires large sample volumes or a high sequencing depth.

Normally, we define the level of gene expression by measuring the reads of mRNA, but the study by Esfahani et al. has given us a new idea. They described promoter fragmentation entropy (PFE) as an epigenome cfDNA feature that predicts RNA expression levels by individual genes and developed a method that uses targeted sequencing of promoters of genes to infer the expression of the targeted TSS, called EPIC-seq [36]. The main reason is that cfDNA fragments near the promoter of expressed genes are more susceptible to cleavage by nucleases and, therefore, have a higher degree of fragmentation [24,75]. In fact, in 2021, the chromatin immunoprecipitation sequencing of cfDNA nucleosomes (cfCHIP-seq) outlined the epigenome distribution of modifications associated with transcriptionally active promoters, enhancers, and gene bodies that are related to gene expression regulation, demonstrating that plasma nucleosomes retain epigenetic information regarding their cell of origin [76]. The degree of fragmentation of cfDNA at promoters correlates with the expression levels of the corresponding genes. However, more efficient methods or tools need to be developed to characterize the expression levels of genes from cfDNA fragmentation.

### 4.3. Identification Tissue of Origin by cfDNA Fragmentation Pattern

cfDNA consists of DNA fragments released from apoptotic and necrotic cells of different tissues of origin and carries the molecular characteristics of the origin cells. Therefore, the tissue of origin can be inferred from cfDNA fragmentomics (Figure 2). The fragment size [39], fragment size distribution [25], preferred end coordinates [29], and jagged ends [58] of cfDNA are important characteristics that contribute to distinguishing individuals with cancer from normal individuals (Table 1). However, these characteristics do not reflect the specific epigenetic characteristics that can determine the tissue origin of tumor-derived cfDNA (Table 1). Here, we describe three methods for identifying the tissue of origin using cfDNA fragmentation.

Girirajan et al. calculated nucleosome occupancy using the window protection score (WPS), defined as the number of molecules spanning the window minus those with endpoints within the window. The tissue origin of cfDNA can be inferred due to the different nucleosome occupancy patterns in different tissues (Figure 2). They suggest that nucleosome spacing inferred from cfDNA in healthy individuals correlates most strongly with the epigenetic features of lymphoid and myeloid cells, and short cell-free DNA fragments reflect the transcription factor footprints [24]. However, the WPS-based method does not characterize the proportion of contributions from different tissues.

In addition to using nucleosome footprints to infer tissue origin, Sun et al. used orientation-aware cfDNA to infer open chromatin regions that could inform the tissue of origin. In open chromatin regions, the end signals of cfDNA can show phase differences due to fragmentation patterns, which refers to differences in the read densities of the upstream (U) and downstream (D) ends of cfDNA molecules in the reference genome (Figure 2). Orientation-aware cfDNA fragmentation (OCF) values are based on the differences in U and D end signals in the center of the relevant open chromatin regions. OCF-based analysis can measure the contribution of cfDNA from different cell types [34].

Liu et al. proposed a method called “FREE-C”, which is defined as the highly correlated fragment lengths between two regions, to infer chromatin organization based on the co-fragmentation pattern of cfDNA. The inferred chromatin compartments reflect different genomic organizations in the various cell types contributing to cfDNA, allowing for the identification and quantification of the tissues of origin [35].

All three methods above use cfDNA fragmentation patterns to infer the characteristics of the epigenome to determine its tissue origin. Moreover, studies have shown that the end motifs of cfDNA fragments differ in tissues and have the potential to reveal the tissue of origin [28].

## 5. Discussion and Prospects

cfDNA fragments are the DNA molecules released via apoptotic or necrotic events and active release mechanisms after enzymatic processing. The fragment length distribution peaks at 166 bp, which is approximately the length of the DNA wrapped by nucleosomes (~147 bp) plus the length of the linker fragment (~20 bp) [77]. The cfDNA from tumor and placental cells is just the length of nucleosome-wrapped DNA. Therefore, the position of nuclease cleavage is not random. In addition to the fragment length of cfDNA, the formation of end characteristics of cfDNA are inseparably related to specific nucleases. DNASE1L3 nuclease deficiency results in aberrations in cfDNA fragment size and a reduction in the “CC” motif of plasma DNA [78]. Consequently, clarifying the intrinsic mechanism of cfDNA formation and clearance by nucleases is biologically important. Currently, few studies on nucleases suggest that cfDNA generation involves DFFB and DNASE1L3, and cfDNA clearance involves DNASE1L3 and DNASE1 [27]. Moreover, the abnormal clearance of cfDNA is associated with SLE. Hence, more emphasis should be placed on the mechanism of cfDNA released from apoptotic cells and the cleavage mechanism of nucleases, which helps us to understand the differences in the cfDNA fragmentomics more thoroughly.

The cfDNA preserved in vitro is subject to degradation and chemical damage [79]. The observed results in vitro may differ from the characteristics of cfDNA in vivo. Therefore, it is still unknown whether pre-analytical factors, including sample physiological conditions, sample collection, sample storage, plasma centrifugation, cfDNA extraction, and sequencing library kits, have an impact on the characterization of fragmentomics. There have been studies comparing different cfDNA extraction kits for cfDNA extraction efficiency and cfDNA integrity. Previous research has characterized the impact of separation time and repeated freezing on cfDNA size using PCR [80]. In addition, the effect of pre-analytical factors on cfDNA fragment length and end motifs has also been investigated. Different stabilizing collection tubes and processing times do not affect the cfDNA fragment sizes, but could impact the genome-wide fragmentation patterns and fragment end sequences of cfDNA [81]. Several studies have been conducted to investigate the effect of pre-analytical factors on cfDNA in order to set a standard processing workflow before it is applied to clinical trials [82,83,84].

Research on cfDNA has focused more on its biological significance, as well as its clinical value. The fragment characteristics of cfDNA can be used as important markers to classify diseases. However, the classification sensitivity does not yet meet the requirements for clinical use. There have been studies on the early diagnosis of cancer by integrating cfDNA fragmentomics characteristics using multiple machine learning models, and good classification results have been obtained [31,85,86]. However, the classification sensitivity is not robust to use with different datasets. Zhang et al. developed a python package (cfDNApipe) for the preprocessing and analysis of cfDNA WGS data [87], but it is not a specific analysis tool for cfDNA fragmentomics characterization. Therefore, a unified software or platform for cfDNA fragmentomics analysis needs to be developed, which would be greatly convenient for liquid biopsy biomarker mining.

cfDNA fragmentomics is the category of histology formed by integrating the fragment characteristics of cfDNA. CfDNA fragments from the genome carry information about the epigenome, which can reflect the tissue of origin [24]. Identifying the tissue origin of cfDNA can help determine the site of pathology, which leads to a more comprehensive understanding of disease diagnosis [22]. In addition, the fragmentomics of cfDNA can noninvasively reflect epigenetic regulation [88]. However, the algorithms for predicting the epigenome using cfDNA fragmentomics are relatively limited, with different resolutions, especially for the prediction of the spatial models of three-dimensional chromosomes. Thus, there is still a long way to go from cfDNA fragmentomics to epigenomics, and more efforts need to be devoted.

In summary, cfDNA analysis for clinical diagnostic applications still faces enormous challenges. One the one hand, cfDNA fragments are susceptible to chemical damage with low concentrations. Hence, many methodological and pre-analytical factors limit the clinical sensitivity of cfDNA-based liquid biopsy cancer detection [89]. On the other hand, more cfDNA fragment characteristics need to be mined, and more bioinformatics methods and tools need to be developed to improve the sensitivity of the detection. In addition to this, higher accuracy in the early screening of cancer and abnormal pregnancy could be achieved by the combination of cfDNA multi-omics, such as fragmentomics and epigenetics. In general, the fragmentomics of cfDNA holds promise for the noninvasive detection of tissue-specific diseases.

## Figures and Tables

**Figure 1 ijms-24-01503-f001:**
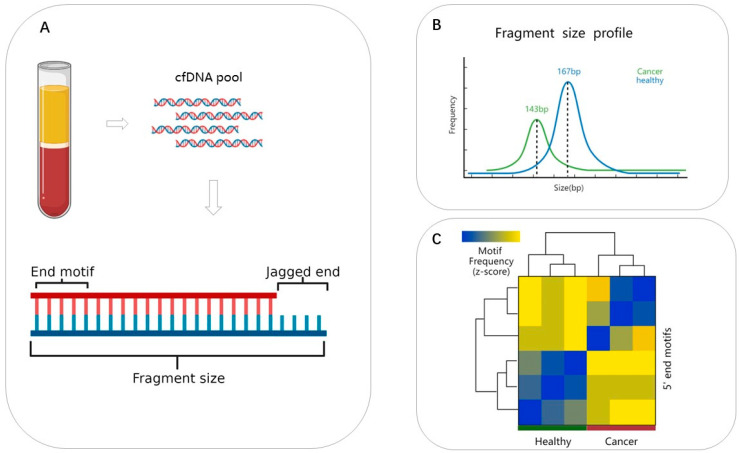
The characteristics of cfDNA differ in healthy individuals and individuals with cancer. (**A**) (**Top**) Different cells, including cancer cells, placental cells in pregnancy, and blood cells, contribute a cell-free DNA pool in the plasma. (**Bottom**) Sketch of cfDNA structure with fragment length, jagged ends and 4-mer end-motifs. (**B**) Overall fragment size distribution of cfDNA in cancer and normal populations. The modal size of normal individuals is 167 bp, and that of individuals with cancer is 143 bp. (**C**) Heat map analysis of 4-mer end motif frequencies between cancer and healthy individuals, which shows the end motifs of cfDNA differ in the two groups.

**Figure 2 ijms-24-01503-f002:**
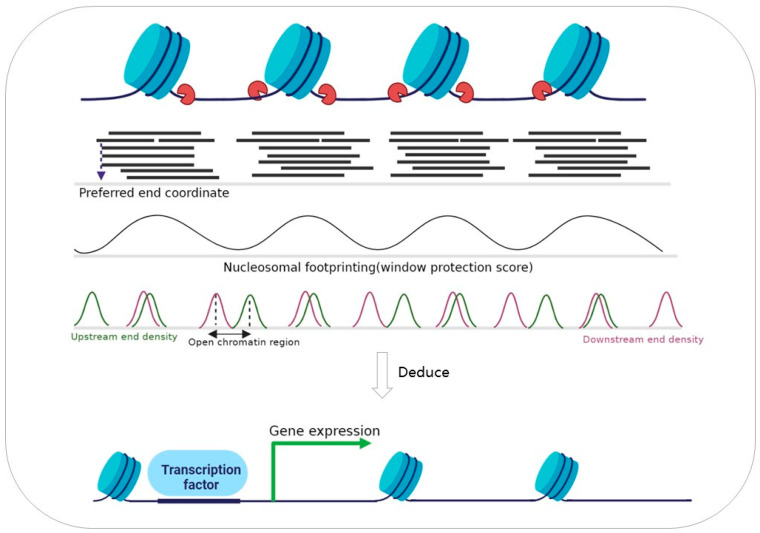
cfDNA fragmentation pattern reflects the epigenetic regulation. The break coordinates of cfDNA tend to cluster at certain locations on the genome, called preferred end coordinates, whose upstream and downstream density can reflect the accessibility of chromatin. Fragments of cfDNA are unevenly distributed in the genome due to the histone protection, which can reveal the tissue-specific nucleosome positioning, transcription factors binding site, and gene expression. In addition, nucleosome footprints are variable across different tissues, thus cfDNA fragmentation patterns convey information regarding the tissue of origin.

**Table 1 ijms-24-01503-t001:** Summary of important cfDNA fragmentomics methods.

Method	Characteristics	Application	Tissue-of-Origin Inference
Calculation of tumor fraction by enriching short fragments [33]	Fragment size	Pan-cancer diagnosis	\
Motif diversity score (MDS) [28]	End motif	Prediction of cancer, pregnancy, and transplantation	Estimation of cell types according to end motif differences
Orientation-aware cfDNA fragmentation (OCF) [34]	Fragment end signals	Diagnosis of pregnancy, liver transplantation, HCC, and CRC	Measurement of the relative contributions of various tissues
Windowed protection score (WPS) [24]	Nucleosome occupancy by cfDNA fragmentation	Monitoring of a set of clinical conditions	Inference of several cell types
DNA evaluation of fragments for early interception (DELFI) [25]	cfDNA fragmentation	Screening, early detection, and monitoring of human cancer	Identifies the tissue of origin of the cancers to a limited number of sites
Fragmentation evaluation of epigenetics from cfDNA (FREE-C) [35]	cfDNA co-fragmentation patterns	Monitoring of in vivo genome organization and quantification of cell death	Quantification of contribution from different cell types
Epigenetic expression inference from cell-free DNA-sequencing (EPIC-seq) [36]	Promoter fragmentation entropy	Prediction of expression for individual genes	Characterization of tissue origin with potential

## Data Availability

Not applicable.

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
