# Peer review of "Cell-Free DNA Fragmentomics: The Novel Promising Biomarker"

_ijms, 2023, doi:10.3390/ijms24021503_

Round 1
Reviewer 1 Report
The review article about “Cell-free DNA Fragmentomics: The Novel Promising Biomarker” gives a global overview about the current research in the field. The article cites the relevant literature and provides a brief outlook. Moreover, two intuitive figures were created visualizing cfDNA characteristics and fragementation patterns.
However, there are some issues that need to be addressed in order to improve the quality of the manuscript. Importantly, several wording and syntax errors are made throughout the paper and need to be corrected. Below are listed some prominent errors. However, preferably a native speaker should proof-read the manuscript.
Line 9: The authors indicate that cfDNA is released via apoptotic or necrotric events (Abstract and Discussion). However, evidence suggests, that active release mechanisms relevantly contribute to the pool of cfDNA. This information need to be implemented in the article, citing relevant literature.
Line 12: Reword “Recent years, it is proved” (for example “Evidence from the recent years…”)
Line 14: change fragment to fragments
Line 34: Change disease to diseases
Line 35: reword “due to unable to”
Line 43-46: The message is not clear to the reader. Is there a high probability of false negatives, due to the high background of cfDNA from heamatopoietic cells? Please clarify.
Line 61: include “that” after shown.
Line 68: Reword “In the late two years” .. does it mean in the last two years?
Line 68: change integrating with integrated
Line 86: Reword “we conclude the approaches”
Line 105: Consider to delete the Initials throughout the text. “Yu et al., …” In Line with this delete “Y M Dennis” Lines 131, 162, 171. Line 241 (Ash A. A.).
Line 125: Include “this technology could be “ before applied.
Line 135: Moulier was not the first author of this article. Change to Mouliers workung group or similar.
Line 141: Sarah is the first name of the author. Please change to lastname. The same accounts for Jiang et al. (Line 156). Please check throughout the manuscript.
Line 200: change wrapped in with wrapped around.
Line 204: change differs to differ
Line 233: change yang to Yang.
Line 274: reword “could informs”
Line 283: Change reflects with reflect
Line 306-310: The fact that different DNA extraction kits (for example Qiagen blood vs. Qiagen cfDNA isolation kit) have large influence on the results is clear. For example PMID: 36417505. Please extend this topic and consider to include it into a paragraph 4.4.
Reviewer 2 Report
Cell-free DNA molecules are released into the plasma via apoptotic or necrotic events, which carrying the genetic and epigenetic information of its origin tissues present a golden opportunity for diagnostics.
However, cfDNA is the mixture of various cell fragments and efficient enrichment of cfDNA fragments with diagnostic value remains a great challenge.
The review highlights that cfDNA fragmentomics characteristics differ in normal and diseased individuals without the need to distinguish the source of the cfDNA fragment, which makes it a promising novel biomarker.
That is , probing the fragment characteristics of cfDNA at the genome-wide level can provide a more effective approach for biomarker mining,
Furthermore, cfDNA fragmentomics can identify tissue origins by inferring epigenetic information. Thus, further insights into the fragmentomics of plasma cfDNA will shed light on the origin and fragmentation mechanisms of cfDNA during physiological and pathological processes in diseases .
The review highlights that enhance our ability to exploit plasma cfDNA as a molecular diagnostic tool.
They focus on the cfDNA fragment characteristics and its potential application, such as fragment length, end motifs, jagged ends, preferred end coordinates, as well as nucleosome footprints, open chromatin region, and gene expression inferred by the cfDNA fragmentation pattern across the genome. The authors furthermore summarize the methods for deducing the tissue-of-origin by cfDNA fragmentomics.
Minor Comments :
- Line 11 Clinical should read clinical setting.
-in vivo and in vitro should be in italics across the manuscript
Overall the review is well written and informative in discsussing fragmentomics of cfDNA and their promise for non-invasive detection of tissue-specific diseases and should be accepted for publication.
